METHODS

# Capturing cell heterogeneity in representations of cell populations for image-based profiling using contrastive learning

**Robert van Dijk**[1], **John Arevalo**[2], **Mehrtash Babadi**[3], **Anne E. Carpenter**[2], **Shantanu Singh**[2]*

**1** CellVoyant Technologies, Bristol, United Kingdom, **2** Imaging Platform, Broad Institute of MIT and Harvard, Cambridge, Massachusetts, United States of America, **3** Data Sciences Platform, Broad Institute of MIT and Harvard, Cambridge, Massachusetts, United States of America

* shantanu@broadinstitute.org

**Data Availability Statement:** All the corresponding data is available as part of the cpg0001-cellpainting-protocol and cpg0004-lincs datasets,

## Abstract

Image-based cell profiling is a powerful tool that compares perturbed cell populations by measuring thousands of single-cell features and summarizing them into profiles. Typically a sample is represented by averaging across cells, but this fails to capture the heterogeneity within cell populations. We introduce CytoSummaryNet: a Deep Sets-based approach that improves mechanism of action prediction by 30–68% in mean average precision compared to average profiling on a public dataset. CytoSummaryNet uses self-supervised contrastive learning in a multiple-instance learning framework, providing an easier-to-apply method for aggregating single-cell feature data than previously published strategies. Interpretability analysis suggests that the model achieves this improvement by downweighting small mitotic cells or those with debris and prioritizing large uncrowded cells. The approach requires only perturbation labels for training, which are readily available in all cell profiling datasets. Cyto-SummaryNet offers a straightforward post-processing step for single-cell profiles that can significantly boost retrieval performance on image-based profiling datasets.

## Author summary

Image-based cell profiling experiments measure thousands of features in millions of cells in microscopy images. Measuring cell response to various treatments in this way has proven powerful for many applications in biological research and drug discovery. However, each cell population responding to a treatment is usually population-averaged, losing information about the diversity of cells within each sample.

We developed CytoSummaryNet, a machine learning method that substantially improves the ability to predict a treatment's mechanism of action from cell imaging profiles. Rather than averaging all cells equally, it learns which cells are most informative; it appears to focus on large, uncluttered cells while downweighting dividing cells and debris.

Training CytoSummaryNet only requires information about which samples were exposed to which treatments, making it simpler to implement than previous techniques.

available from the Cell Painting Gallery on the Registry of Open Data on AWS (https://registry.opendata.aws/cellpainting-gallery/), and released under CC0 1.0 Universal license. **Code availability** All code to reproduce this analysis is available at https://github.com/carpenter-singh-lab/2023_vanDijk_CytoSummaryNet.

**Funding:** Funding for the project was provided by the National Institutes of Health NIGMS (R35 GM122547 to AEC) and an internship funded by the Massachusetts Life Sciences Center, Stichting dr. Hendrik Muller's Vaderlandsch Fonds, Stichting de Fundatie van de Vrijvrouwe van Renswoude te 's-Gravenhage, and Fund International Experience/ Holland Scholarship (to RvD). The funders had no role in study design, data collection and analysis, decision to publish, or preparation of the manuscript.

**Competing interests:** S.S. and A.E.C. serve as scientific advisors for companies that use image-based profiling and Cell Painting (A.E.C: Recursion,SyzOnc, Quiver Bioscience; S.S.: Waypoint Bio,Dewpoint Therapeutics, Deepcell) and receivehonoraria for occasional talks at pharmaceutical and biotechnology companies. R.v. D is an employee of CellVoyant. All other authors declare no competing interests.

By learning biologically relevant ways to summarize single-cell data into treatment profiles, CytoSummaryNet could help scientists leverage cell heterogeneity to extract richer insights from cell imaging screens and ultimately accelerate the development of new therapies.

## Introduction

High-throughput assays enable quantifying cellular responses to perturbations at a large scale. Image-based assays are among the most accessible and inexpensive such technologies that offer single-cell resolution. Cell populations are perturbed with compounds or genetic perturbations, stained, and then imaged. Large amounts of quantitative morphological data are extracted from these microscopy images, generating tabular data comprising single cell profiles. These single cell profiles are then aggregated by a variety of methods to a per-perturbation profile that describes that population's phenotype. The profiles of different cell populations can be compared to identify previously unrecognized cell states induced by experimental perturbations. This method, called image-based cell profiling, is a powerful tool that can be used for drug discovery, functional genomics, and disease phenotyping [1]. Among other applications, image-based profiling has already been used to predict assay outcomes for compounds [2–4], detect leukemia label-free [5], and predict the impact of particular gene mutations [6].

Image-based cell profiling shows great potential, but many steps in its pipeline can still be improved [1]. One of the challenges is to capture both population trends and single-cell variability. Cell populations are known to be heterogeneous [7,8], and recent studies have yielded many insights into the mechanisms and importance of this characteristic [9–12]. Capturing that heterogeneity could improve a profile's information content and, thus, utility in many image-based profiling applications.

Despite its current limitations, so-called population-averaged profiling, where all single-cell features are averaged per feature using either the mean or the median, remains the most commonly-used approach in the field of image-based profiling. This is true regardless of the types of features or the kinds of transformations (e.g., normalization, feature selection, etc.) applied to the profiles [13,14]. Average profiling is a simple way of summarizing a cell population (hereafter referred to as a sample) into a vector (a sample's profile) with only one value per measured feature. It dramatically decreases the data size (as there are typically thousands of cells per well, hundreds of wells per plate, and multiple plates per experiment) and simplifies downstream analysis.

The biggest drawback of average profiling is that information on cell subpopulations is lost. This can result in identical average-aggregated profiles despite cell populations having distinct internal structures. Additionally, ignoring subpopulations can lead to a quantitatively incorrect interpretation. For example, two cell populations can show correlations among certain features when averaged but show completely different relations when compared after grouping the cells, i.e., Simpson's paradox [15]. Lastly, averaging a sample essentially assumes each measured feature corresponds to a simple unimodal distribution. If this is not the case, e.g., as in the case of heterogeneous cell populations, the average will be a poor summary statistic for the data.

Several methods have been proposed to capture the heterogeneity of cell populations into their corresponding profiles. The most straightforward solution is to incorporate the cell population's dispersion (e.g., standard deviation) for each extracted feature and concatenate these values with the average-aggregated profile. This approach is widely adopted but offers only

minor improvements over average profiling [16]. A later study suggested that incorporating the sample histogram–instead of dispersion–may offer improvements [17]. A different approach involves first clustering cells and then calculating the profiles based on their subpopulations [18,19]. These methods capture more information about subpopulations rather than only incorporating their dispersions but did not significantly improve upon average profiling [16]. Furthermore, they can lead to incomparable profiles across experiments unless the subpopulations are defined beforehand. As well, many cell phenotypes are better described with a continuous rather than a discrete scale [13].

In our previous work, we improved the performance of average profiling by fusing features' averages, dispersion, and covariances [13]. This method provided ~20% better performance predicting a compound's mechanism of action and a gene's pathway, showing that capturing statistics related to cell population heterogeneity can improve performance on downstream tasks. However, this method has two major limitations. First, it only captures the first- and second-order moments of the data. Second, because it produces a similarity matrix rather than an embedding, it requires recomputing the pairwise similarities among all profiles each time a new profile is included in the dataset. Here, we introduce a novel method that addresses both of these limitations and automatically finds an effective way to aggregate single-cell data to improve the information content of sample profiles.

## Results

### Experimental design and methodology for learning cell representations

We propose CytoSummaryNet: a self-supervised contrastive learning approach that leverages the naturally available information in profiling experiments (Fig 1B). Contrastive learning is a method where data points corresponding to the same entity–for example the same sample–are brought closer together while others are pushed apart in a feature space [20]. In doing so, it aims to capture generalizable features beneficial for a broad range of tasks. Here, we use perturbation identifications (IDs) as labels to train a latent feature space that distinguishes samples with different IDs. In this feature space, profiles of replicates with the same perturbation should be close to each other, while those of different perturbations should be further apart (Fig 1D).

This labeling approach frames the problem as a multiple-instance learning problem [22], assuming that the replicate wells consist of cells with similar feature distributions, and that different compounds generate populations with distinct feature distributions. Although not every perturbation yields a profile distinct from all others, these assumptions collectively contribute to the development of a feature embedding that captures biologically significant morphological variations. Importantly, we here apply contrastive learning to already-extracted single-cell features, as opposed to raw pixels, making it immediately applicable to most image-based profiling datasets, where extracted features are readily available.

We treat the data as a collection of sets of cells, where each sample (all cells from a single well) corresponds to one set (based on the mathematical definition of "set"). To aggregate cells from such a sample into a profile, the function we want to learn should possess a few properties. First, it should be capable of handling input samples of arbitrary sizes. Second, because cells within a sample are, by definition, unordered, the function should be permutation invariant. Several methods have been developed for analyzing this type of data [23,24], and a general formulation for addressing this problem is known as Deep Sets [25]. Zaheer et al. [25] revealed that a function acting as a universal approximator for sets has a specific structure, consisting of a permutation-invariant function and a learnable representation function. This structure provides a foundation for designing neural networks capable of processing unordered data represented as sets (Fig 1C).

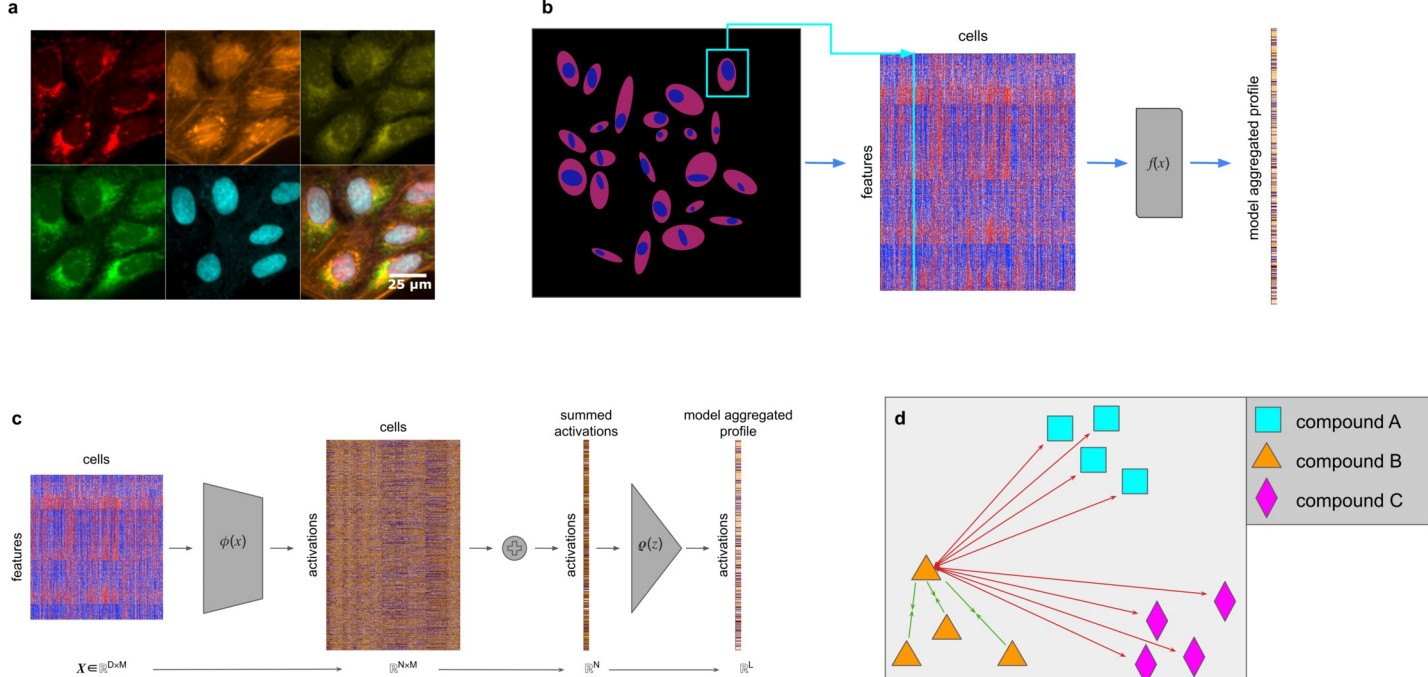

**Fig 1.** (a) Human U2OS cells treated with dimethyl sulfoxide (DMSO) and stained using the Cell Painting assay, which employs six dyes in five channels to label eight cellular compartments. The top row (from left to right) shows mitochondrial staining; actin, Golgi, and plasma membrane staining; and nucleolar and cytoplasmic RNA staining. The bottom row (from left to right) displays endoplasmic reticulum staining, DNA staining, and a montage of all five channels (from Cimini et al. [21]). (b) Thousands of features are extracted from each segmented cell in microscopy images of wells. A learned function $f(x)$ (CytoSummaryNet) aggregates this data into a single feature vector: the sample's profile. (c) An in-depth look at the model architecture used in this study. The model consists of three elements: a function $\varphi(x)$, which maps the input data from $\mathbb{R}^D$ to $\mathbb{R}^L$ space, a summation, which collapses the cell dimension, and $\rho(z)$, which maps the collapsed representation from $\mathbb{R}^N$ to $\mathbb{R}^L$ space. (d) During training, replicate compound profiles are forced to attract each other (green arrows) and simultaneously repel every other compound (red arrows) in the learned feature space. Here, all forces are drawn for a single profile of compound B.

We therefore chose to employ the Deep Sets formulation to learn a model (CytoSummaryNet) that aggregates single-cell feature data into a profile (CytoSummaryNet profiles) that outperforms population-averaged profiles in predicting a compound's mechanism of action. This is achieved through self-supervised contrastive learning within a multiple-instance learning framework, which allows the model to process groups of data–labeled only at the group level rather than the individual data point level–and to make classifications based on the collective information of these groups. We evaluate the training task (replicate retrieval, i.e., retrieving profiles of other replicate wells of the same compound) and the downstream task (retrieving profiles of other compounds annotated with the same mechanism of action) using the mean average precision (mAP) metric [26]. This information retrieval metric indicates whether all the positive examples can correctly be identified without erroneously marking too many negative examples as positive. We evaluated CytoSummaryNet on two fronts: (i) testing its generalizability across unseen compounds and experimental protocols, and (ii) a practical application scenario where we trained the model on a large dataset and subsequently measured its performance on mechanism of action retrieval for that dataset.

## CytoSummaryNet learns embeddings that generalize to unseen compounds

We first investigated CytoSummaryNet's capacity to generalize to out-of-distribution data: unseen compounds, unseen experimental protocols, and unseen batches. The data split strategy is

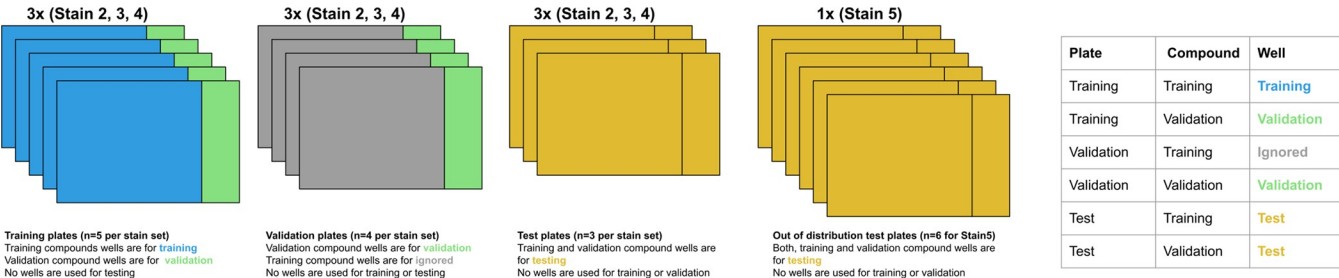

**Fig 2. Stratification of the cpg0001 dataset.** The dataset is divided into four subsets–Stain2, Stain3, Stain4, and Stain5 –each corresponding to a specific set of assay conditions designed to optimize Cell Painting. Stain2, Stain3, and Stain4 contain training, validation, and test plates, while Stain5 consists solely of test plates, serving as an out-of-distribution test set with experimental conditions entirely different from the other subsets. Within Stain2 and Stain3, each plate had slight variations in assay conditions, resulting in strong batch effects. Although addressing batch or experiment effects is not the primary focus of this study, test set plates were deliberately selected to represent the most divergent conditions within Stain2, Stain3, and Stain4, ensuring their out-of-distribution nature. The dissimilarity between the test plates and the training and validation data was used as the basis for selecting the test plates for Stain2, Stain3, and Stain4. Fig E in S1 Text elaborates on the method used to measure this similarity, Table C in S1 Text provides the plate names for each dataset in this stratification, and Fig H in S1 Text describes the training and validation compound split for all plates.

visualized in Fig 2, and the results of these investigations are visualized in Figs 3, 4, and 5, respectively. We performed this analysis on the cpg0001 dataset [21] from the public Cell Painting Gallery [27], which consists of 384-well plates with identical sample layouts of 90 unique compounds, i.e., there is a single "plate layout" for the entire dataset (See *Experimental Setup*: *Data*); each plate contains four replicates of each compound in different well positions, plus 24 negative controls. The dataset consists of many experimental plates, most with different experimental conditions to optimize the image-based cell profiling protocol. The dataset is categorized into four subsets of plates: Stain2, Stain3, Stain4, and Stain5. Each of these subsets corresponds to a specific set of assay conditions that were designed to optimize Cell Painting [21,28,29]–the most widely used image-based profiling assay [30]–for detecting morphological phenotypes and grouping similar

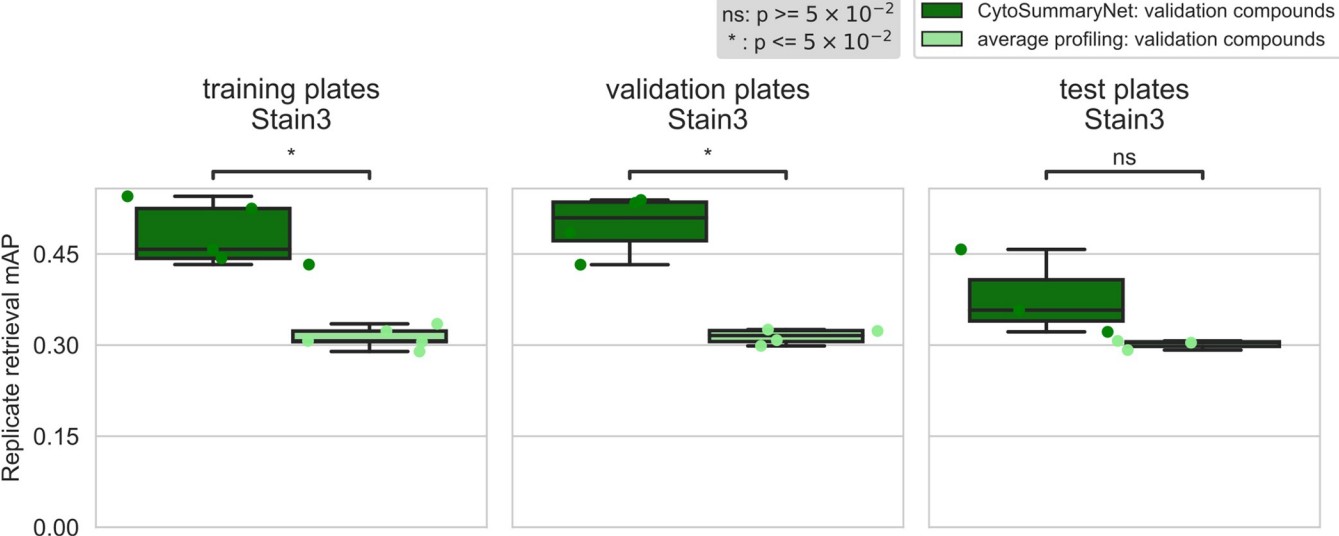

**Fig 3. CytoSummaryNet profiles generally outperform average-aggregated profiles for sensitively identifying replicates of a given sample, and partially generalize to unseen experimental protocols (test plates).** The box plots illustrate the mAP of replicate retrieval for all validation compounds of Stain3 (each data point is the average mAP of a plate) by CytoSummaryNet (dark green) and average (light green) profiles. Note that although the panels are labeled "training plates" and "validation plates", all data shown comes from validation **compounds** and therefore none of it has been directly seen during training (see description of stratification in Results for further details). Welch's t-tests were used to compare the means between CytoSummaryNet and average mAP scores on corresponding data; their p-values are indicated as stars at the top of each plot (ns = not significant).

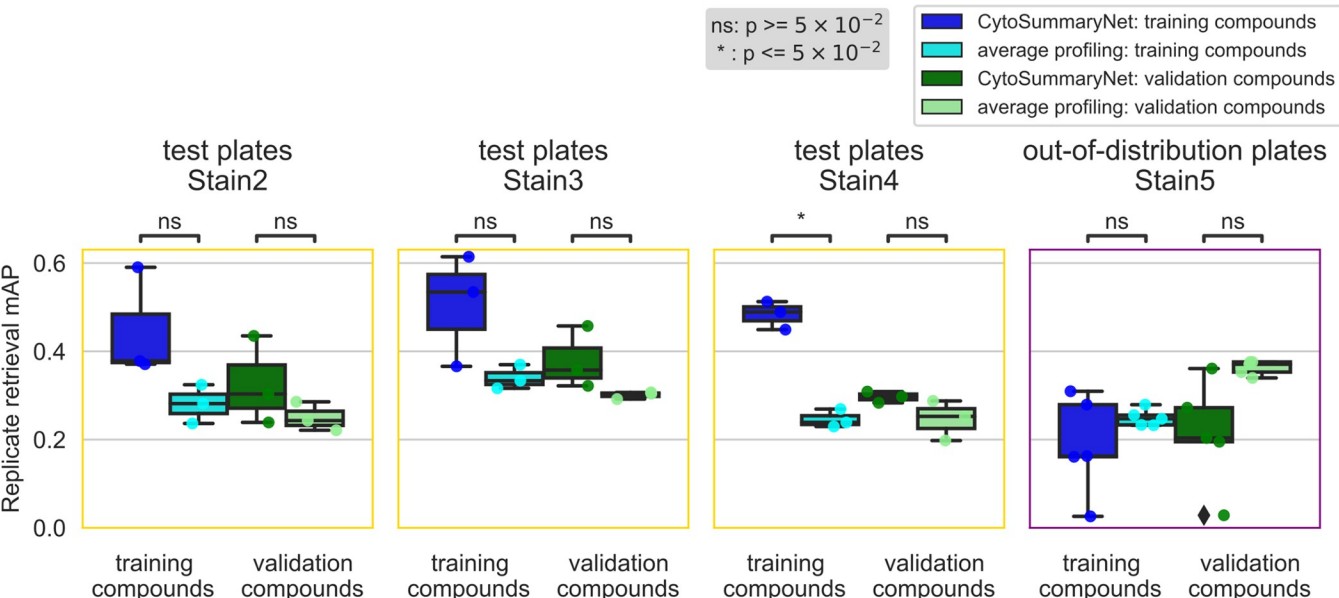

**Fig 4. CytoSummaryNet profiles partially generalize to unseen compounds and do not generalize to out-of-distribution batch data (Stain 5).** The box plots illustrate the mAP of replicate retrieval for training and validation compounds (each data point is the average mAP of a plate) from the test plates of Stain2, Stain3, Stain4, and Stain5; CytoSummaryNet profiles performance in dark blue and dark green respectively, and average profiles performance in cyan and light green respectively. Note (i) the boxplots corresponding to validation compounds in the second panel ("test plates Stain3") are the same as the boxplots in the third panel of Fig 3, and (ii) although the boxes are labeled "training compounds" and "validation compounds", all data shown comes from test plates and therefore none of it has been seen during training (see description of stratification in Results for further details). Welch's t-tests were used to compare the means between CytoSummaryNet and average mAP scores on corresponding data; their p-values are indicated as stars at the top of each plot (ns = not significant). The limited number of data points, due to averaging mAP scores per plate, may impact the statistical significance of the comparisons.

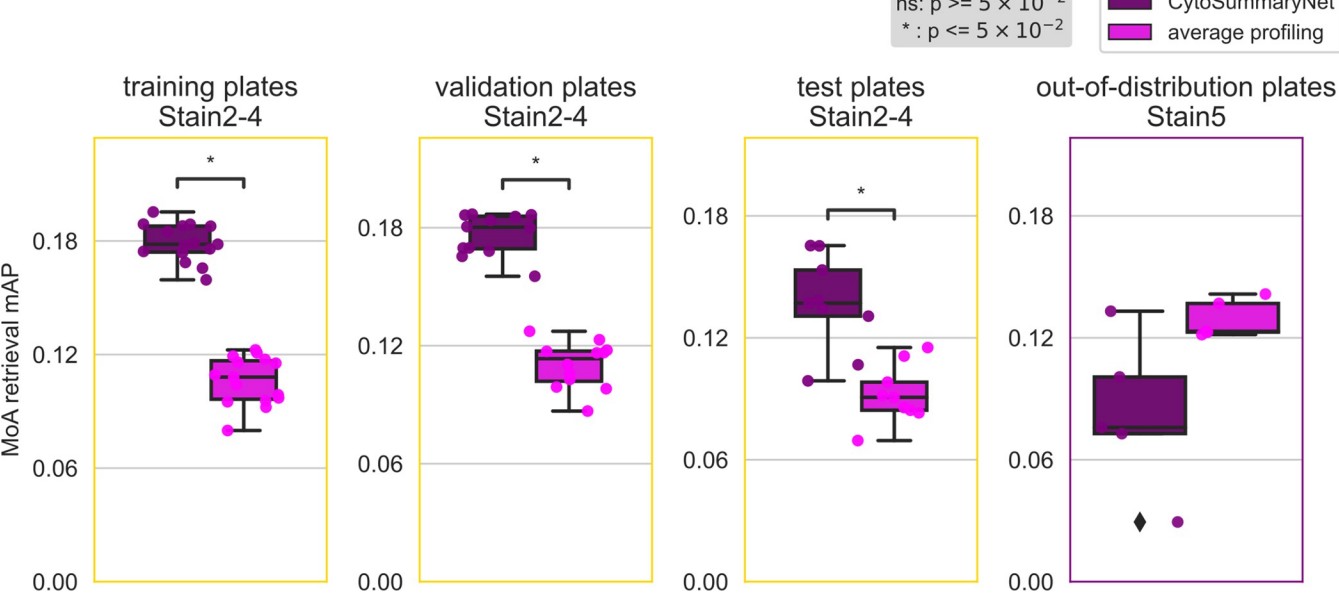

**Fig 5. CytoSummaryNet profiles generally outperform average-aggregated profiles in mechanism of action (MoA) retrieval, although not for out-of-distribution batch data (Stain5).** The box plots illustrate the average mAP of mechanism of action retrieval for CytoSummaryNet- (dark purple) and average- (light pink) aggregated profiles (each data point is the average mAP of a plate). Welch's t-tests were used to compare the means between CytoSummaryNet and average mAP scores; their p-values are indicated as stars inside each plot (ns = not significant). The limited number of data points, due to averaging mAP scores per plate, may impact the statistical significance of the comparisons.

perturbations together (See *Experimental Setup*: *Diversity of stain sets*). The partitioning of the Stain experiments have been defined and explained previously [21].

The data was split into training, validation, and test sets based on two variables: plate and compound (Fig 2). Stain2, Stain3, and Stain4 were divided into train, validation, and test plates, while Stain5 consisted solely of test plates and was considered out-of-distribution. Additionally, 20% of the compounds (n = 18) were randomly selected and designated as validation compounds, with the remaining compounds assigned to the training set (Fig H in S1 Text). The training set, consisting of wells from training plates treated with training compounds, was used to update the model weights. The validation set, composed of wells from training or validation plates treated with validation compounds, was employed to select the optimal model. Finally, the test set, containing wells from test plates, was used for hold-out evaluation. Given the limited number of compounds, this stratification strategy allowed for the evaluation of CytoSummaryNet's performance on 18 compounds that were not encountered during training. The selection of test plates within each subset was based on their dissimilarity to the training and validation data, as described in Fig 2.

Profiles from CytoSummaryNet trained on Stain2, Stain3, and Stain4 consistently showed significantly higher mAP scores for the replicate retrieval task than the baseline average-aggregated profiles, on the training and validation plates (panels 1 and 2 of Fig 3, Stain3; comparable results on Stain2 and Stain4 in *Fig A in S1 Text*). We also saw improvement on the test plates (panel 3 of Fig 3), however, the improvement is not statistically significant, indicating some degree of overfitting due to learning plate specific effects. The improvement CytoSummaryNet provides is diminished overall for held-out validation compounds compared to compounds included in the training set, indicating additional overfitting on the compounds seen during training (Fig 4). The poor performance on Stain5, the subset from which no plates were used for training CytoSummaryNet, suggests that CytoSummaryNet does not generalize to out-of-distribution batches of data that used a significantly different experimental protocol.

## CytoSummaryNet improves mechanism of action prediction compared to average profiling

We next addressed a more challenging task: predicting the mechanism of action class for each compound at the individual well level, rather than simply matching replicates of the exact same compound (Fig 5). It's also important to note that mechanism of action matching is a downstream task on which CytoSummaryNet is not explicitly trained. Consequently, improvements observed on the training and validation plates are more meaningful in this context, unlike in the previous task where only improvements on the test plate were meaningful. For similar reasons, we calculate the mechanism of action retrieval performance on all available compounds, combining both the training and validation sets. This approach is acceptable because we calculate the score on so-called "sister compounds" only—that is, different compounds that have the same mechanism of action annotation. This ensures there is no overlap between the mechanism of action retrieval task and the training task, maintaining the integrity of our evaluation. Table 1 summarizes the results discussed in this and the following section, highlighting the performance differences across various batches.

As expected, the baseline mean Average Precision (mAP) for this task is much lower, due in part to the well-known imperfect annotation of compounds with respect to their mechanism. Despite this, we find that CytoSummaryNet profiles significantly improve the mAP scores for mechanism of action retrieval compared to average-aggregated profiles. There is, however, one exception: Stain5 test plates, which require generalizing to out-of-distribution batch data. In this case, CytoSummaryNet's performance was poor (Fig 5), consistent with the replicate retrieval results (Fig 4).

**Table 1. Top panel: Absolute and relative average improvements in mAP of mechanism of action retrieval between CytoSummaryNet- and average-aggregated profiles for the cpg0001 dataset.** The improvements are calculated as `mAP(CytoSummaryNet)-mAP(average profiling)`. The percentage improvements are calculated as `(mAP(CytoSummaryNet)-mAP(average profiling))/mAP(average profiling)`. Bottom panel: The same mAP improvements but for the cpg0004 dataset, for CytoSummaryNet models that aggregate single-cell information (single-cell) and that transform population average information (population average).

| dataset | stratification | mAP improvement Stain2 | mAP improvement Stain3 | mAP improvement Stain4 | average mAP improvement Stain2, Stain3, and Stain4 | mAP improvement Stain5 |
|---|---|---|---|---|---|---|
| cpg0001 | training | 0.081 (81%) | 0.065 (55%) | 0.073 (69%) | 0.073 (68%) | |
| | validation | 0.068 (70%) | 0.060 (50%) | 0.071 (63%) | 0.066 (61%) | |
| | test | 0.043 (52%) | 0.039 (36%) | 0.052 (60%) | 0.045 (49%) | -0.047 (-36%) |
| | | mAP improvement (single-cell) | mAP improvement (population average) | | | |
| cpg0004 | 10 μM (training/ validation) | 0.021 (68%) | 0.010 (32%) | | | |
| | 3.33 μM (test) | 0.009 (30%) | | | | |

Overall, on the mechanism-retrieval task, the CytoSummaryNet profiles achieve a mAP that is 68%, 61%, and 49% higher than average profiling for all of the training, validation, and test set plates (excluding the out-of-distribution data from Stain5), respectively (Table 1, top panel). Interestingly, average profiling demonstrates superior performance on out-of-distribution Stain5, outperforming CytoSummaryNet with a 36% higher mAP on average.

## Practical application of CytoSummaryNet: Improved mechanism of action retrieval and dose generalization

We tested CytoSummaryNet in a more practical use case: training the model using compound IDs and then using it to infer improved profiles for the mechanism of action retrieval task on the same dataset. Critically, the pretext task used for training—attracting replicate profiles—is entirely distinct from the biologically relevant evaluation task, which is mechanism of action retrieval. This distinction ensures the validity of our approach, despite using the same dataset for both tasks. This mirrors the mechanism of action evaluation discussed in the previous section. For this purpose, we used a different dataset: Batch 1 of the cpg0004 dataset [28], which includes 1,258 unique compounds. We chose the 10 μM dose point for training because we expected this high dosage to produce stronger profiles with more variance than lower dose points, making it more suitable for model training.

The multiple dose points in this dataset allowed us to create a separate hold-out test set using the 3.33 μM dose point data. This approach aimed to evaluate the model's performance on data with potentially weaker profiles and less variance, providing insights into its robustness and ability to capture biologically relevant patterns across dosages. While cross-validation on the 10 μM dose could also be informative, focusing on lower dose points offers a more challenging test of the model's capacity to generalize beyond its training conditions, although we do note that all compounds' phenotypes would likely have been present in the 10 μM training dataset, given the compounds tested are the same in both.

In this context, we found that mechanism of action retrieval was again more successful using CytoSummaryNet profiles compared to the average baseline (Fig 6; most data points are found in the lower right quadrant when using the identity line as a divider). The mean Average Precision (mAP) averaged over all mechanisms of action is 68% and 30% higher when using CytoSummaryNet profiling compared to average profiling for the 10 μM (training/validation) and 3.33 μM (test) dose points, respectively (Table 1, bottom panel).

CytoSummaryNet generally amplifies profiles with mAP > 0.1 in average profiling, where some signal is already present. Interestingly, the points near the origin show that

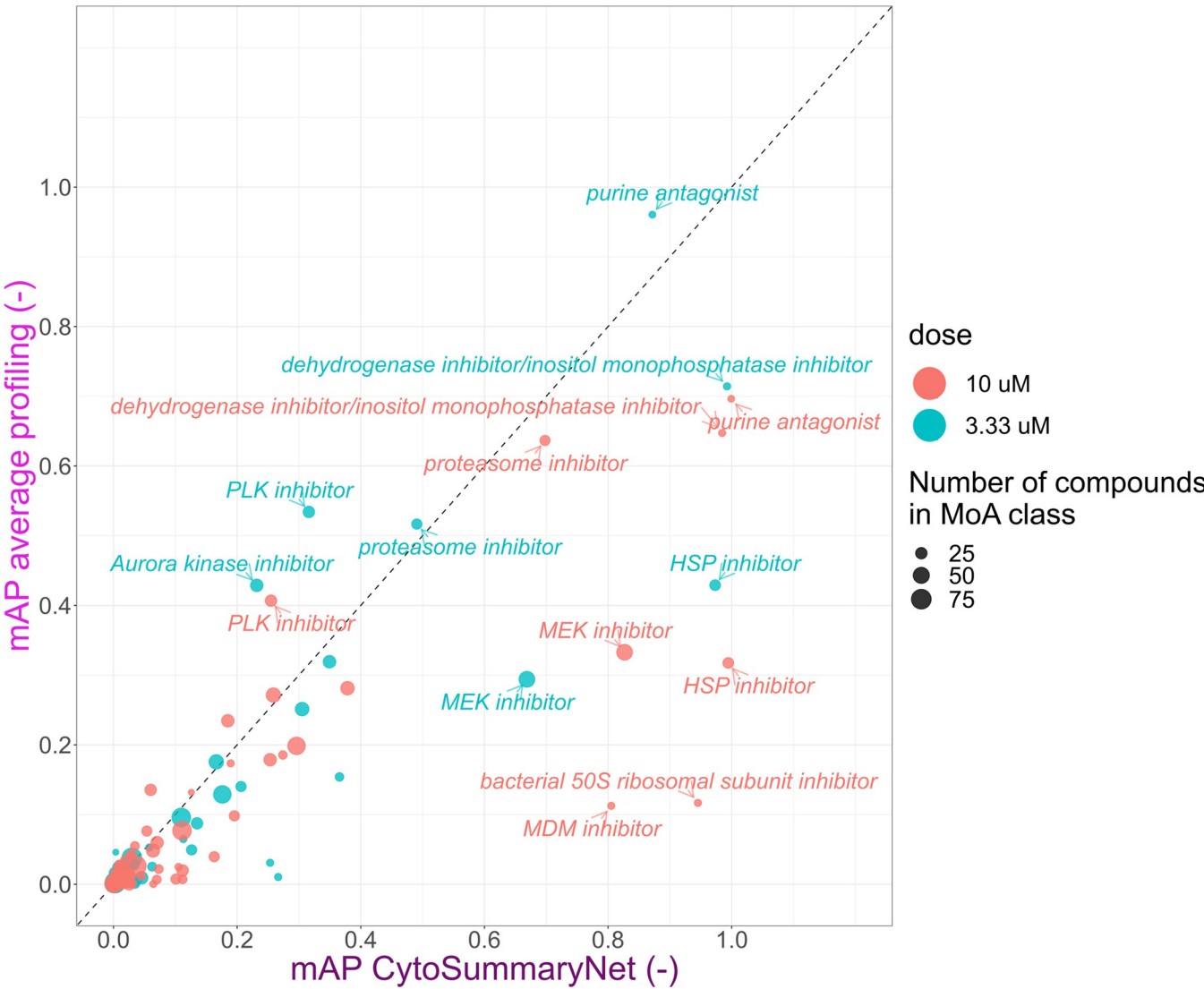

**Fig 6. CytoSummaryNet profiles make readily identifiable mechanisms of action even easier to find and allow for the discovery of previously unfindable mechanisms of action when using average-aggregated profiles (cpg0004 dataset).** Mean average precision (mAP) of average and CytoSummaryNet-based profiling for mechanism of action retrieval of 3.33 μM (test) and 10 μM (training/validation) dose point compound perturbations. We highlight certain high-performing mechanisms of action. Data points are scaled in size based on the number of compounds labeled with that mechanism of action (MoA).

CytoSummaryNet also amplifies the strength of some profiles that were not visible before using average profiling, increasing the mAP from less than 0.05 to more than 0.1.

These results suggest that the use of CytoSummaryNet can make readily identifiable mechanisms of action even easier to detect and allow for the discovery of previously undetectable mechanisms of action. Few CytoSummaryNet profiles achieve a lower mAP than average profiling, suggesting minimal drawbacks to using CytoSummaryNet.

### Contrastive learning alone improves mechanism of action retrieval, but single-cell data boosts performance

We conducted an experiment to determine whether the improvements in mechanism of action retrieval were due solely to CytoSummaryNet's contrastive formulation or also influenced by

the incorporation of single-cell data. We applied the CytoSummaryNet framework to pre-processed average profiles from the 10 μM dose point data of Batch 1 (cpg0004 dataset). This approach isolated the effect of the contrastive architecture by eliminating single-cell data variability. We adjusted the experimental setup by reducing the learning rate by a factor of 100, acknowledging the reduced task complexity. All other parameters remained as described in earlier experiments.

This method yielded a less pronounced but still substantial improvement in mechanism of action retrieval, with an increase of 0.010 (32% enhancement—Table 1). However, this improvement was not as high as when the model processed single-cell level data (68% as noted above). These findings suggest that while CytoSummaryNet's contrastive formulation contributes to performance improvements, the integration of single-cell data plays a critical role in maximizing the efficacy of mechanism of action retrieval.

## CytoSummaryNet captures higher-order moments and enhances cluster separation

We investigated how CytoSummaryNet achieves improved results for replicate and mechanism of action retrieval. As noted in the introduction, previous studies achieved some improved performance by using second-order moments in addition to the average [13]. Our supporting experiments using a toy dataset (*Figs B-D and Tables A and B in S1 Text*) indicate that CytoSummaryNet can learn both second-order moments (covariance) and third-order moments (skewness). We then used uniform manifold approximation and projection (UMAP [31]) and saliency analyses to investigate the features and patterns captured by CytoSummaryNet.

UMAP analysis reveals that CytoSummaryNet profiles better distinguish samples (Fig 7B and 7D) compared to average-aggregated profiles (Fig 7A and 7C). For visualization, we show only plates from Stain3. The UMAP of CytoSummaryNet profiles displays six easily distinguishable clusters of compound profiles, each sharing the same mechanism of action annotation. Remaining clusters beyond the central main cluster are organized based on compound replicates, although not being grouped by mechanism of action. Notably, there are no distinct clusters with mixed mechanisms of action, nor clusters formed according to plates.

In contrast, the UMAP of average-aggregated profiles shows only four easily distinguishable clusters of compound profiles sharing a common mechanism of action (Fig 7A and 7C). Other compound profiles are either part of the main central cluster or one of three smaller clusters in the top-left, bottom, or bottom-right of the plot. The bottom-right cluster contains profiles from a single plate with a significantly different experimental protocol, involving microscopy image pixel binning before extracting single-cell features (Fig 7E). *Fig E in S1 Text* further demonstrates how this plate (BR0015134bin1) differs markedly from other plates in the Stain3 subset using similarity analysis.

## Relevance scores reveal CytoSummaryNet's preference for large, isolated cells

To investigate what kinds of cells are emphasized during CytoSummaryNet aggregation, we calculated a relevance score for each cell. As mentioned previously, the features were extracted using CellProfiler, with a pipeline that measures thousands of single-cell features. The relevance score assesses the importance of each cell in the aggregation process by combining sensitivity analysis (SA) and critical point analysis (CPA). SA evaluates the model's predictions by analyzing the partial derivatives in a localized context, while CPA identifies the input cells with the most significant contribution to the model's output. The relevance scores of SA and CPA

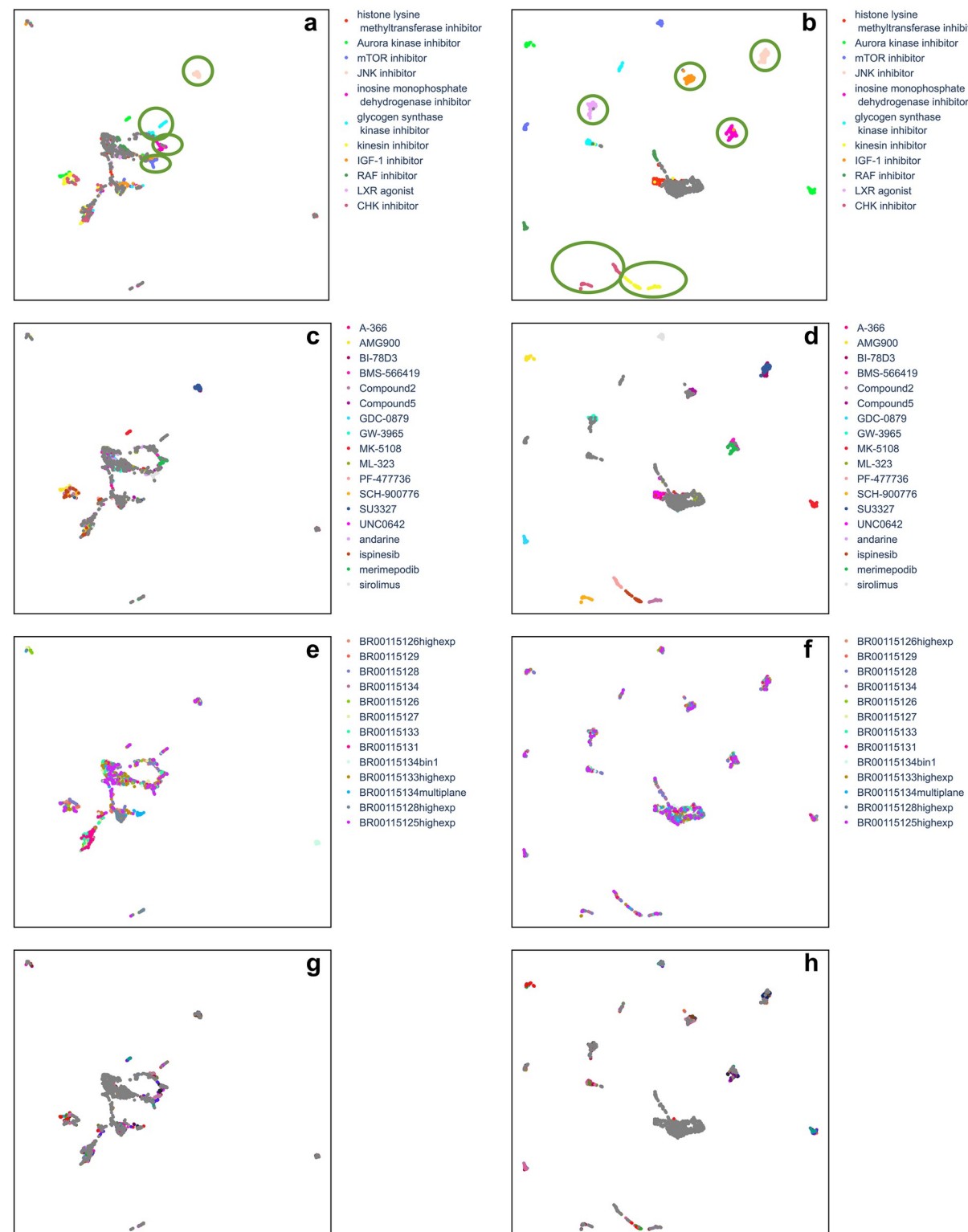

**Fig 7. CytoSummaryNet profiles show a better ability to distinguish similar samples than average-aggregated profiles (cpg0001 dataset).** UMAP of the average- (left column) and CytoSummaryNet- (right column) aggregated profiles of the top 15 mechanisms of action, based on CytoSummaryNet's mAP scores for mechanism of action retrieval, from all used cpg0001 Stain3 plates (Table C in S1 Text). The UMAP was created using *n_neighbors* = 15 and cosine similarity as a distance measure. The profiles are colored based on their corresponding annotated mechanism of action, compound, plate, and well position from top to bottom, respectively. Same mechanism of action profile clusters that

were visible for multiple *n_neighbors* values are annotated in green ellipses, respectively. Note that Compound2 and Compound5 are intentionally anonymized.

are min-max normalized per well and then combined by addition. The combination of the two is again min-max normalized, resulting in the SA and CPA combined relevance score (see *Methods* for details).

The combination of CellProfiler features that most highly correlate with the SA and CPA combined relevance score (Table 2) suggests that features associated with crowded cells (which are smaller and often have elevated DNA signal from adjacent cells measured in the cytoplasm or at the cell edge) are given less weight in the final aggregated profile. In contrast, larger and more isolated cells receive a higher weight. *Tables E-J in S1 Text* list an additional 15 most-correlated features.

To further validate CytoSummaryNet's prioritization of large, uncrowded cells, we progressively filtered cells based on *Cells_AreaShape* features and observed the impact on replicate retrieval mAP (*Fig K in S1 Text*). The results support our interpretation and highlight the key role of larger cells in profile strength.

An examination of an image, selected based on the substantial increase in the corresponding profile's mechanism of action retrieval mAP score when using CytoSummaryNet profiling compared to average profiling (Fig 8), reveals that the most relevant cells are larger, generally isolated from other cells, and do not contain spots of high-intensity pixels (green arrow). Conversely, the least relevant cells display contrasting characteristics; they are more often clumped together (bottom purple arrow) or contain spots of high-intensity pixels (top left and top right purple arrows). These findings are consistent with the conclusions drawn from Table 2, even though they were derived from a different experimental batch (Stain2 instead of Stain3), indicating that emphasizing certain cell types is a general way for CytoSummaryNet to improve upon average profiling.

Statistical t-tests were conducted to identify the features that most effectively differentiate mechanisms of action from negative controls in average profiles, focusing on the three mechanisms of action where CytoSummaryNet demonstrates the most significant improvement and the three mechanisms where it shows the least. Consistent with our hypothesis that CytoSummaryNet emphasizes larger, more sparse cells, the important features for the CytoSummaryNet-improved mechanisms of action (Fig I in S1 Text) often involve the radial distribution for the mitochondria and RNA channels. These metrics capture the fraction of those stains near

**Table 2. Top five CellProfiler features based on their positive and negative Pearson correlation coefficient with the SA and CPA combined relevance score.** The scores were calculated for a single test plate of cpg0001 Stain3 (200922_015124-V).

| Feature category | Feature name | Correlation coefficient |
|---|---|---|
| Cytoplasm | Cytoplasm_Correlation_K_DNA_Brightfield | 0.72 |
| Cells | Cells_AreaShape_MeanRadius | 0.71 |
| Cells | Cells_AreaShape_MaximumRadius | 0.70 |
| Cells | Cells_AreaShape_MedianRadius | 0.70 |
| Cells | Cells_AreaShape_Area | 0.68 |
| Cells | Cells_Intensity_MeanIntensityEdge_DNA | -0.74 |
| Cytoplasm | Cytoplasm_Intensity_MeanIntensityEdge_DNA | -0.72 |
| Cytoplasm | Cytoplasm_Intensity_UpperQuartileIntensity_DNA | -0.71 |
| Cytoplasm | Cytoplasm_Intensity_MeanIntensity_DNA | -0.69 |
| Cytoplasm | Cytoplasm_Correlation_K_Brightfield_DNA | -0.67 |

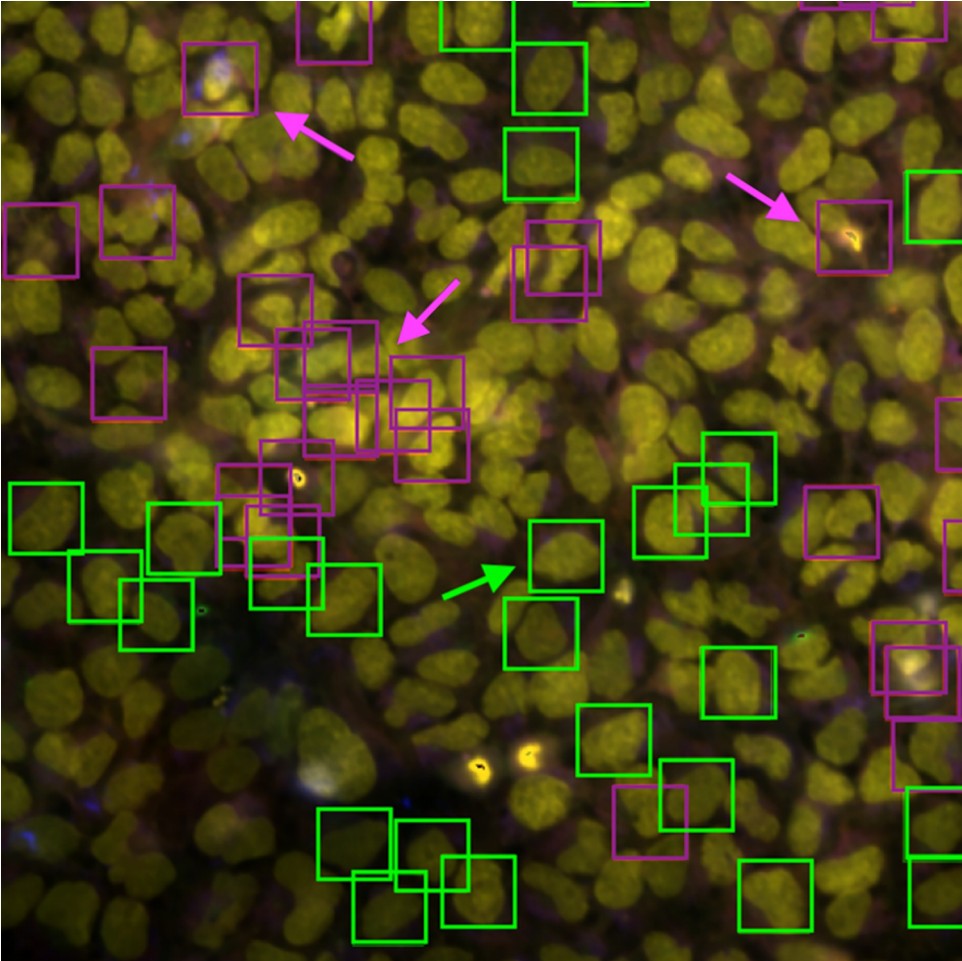

**Fig 8. Five-channel combined microscope image of one of the four fields of view for a well in plate cpg0001 Stain2 BR00112197binned (human U2OS osteosarcoma cells).** The most relevant cells are annotated with green boxes and the least relevant cells are annotated with purple boxes. Three cells characteristic for low relevance scores are explicitly labeled with purple arrows. One cell characteristic for high relevance scores is explicitly labeled with a green arrow. The other three images from this well are shown in Fig F and G in S1 Text.

the edge of the cell versus concentric rings towards the nucleus, which are more readily detectable in larger cells compared to small, rounded cells.

In contrast, the important features for mechanisms of action not improved by CytoSummaryNet (Fig I and Table K in S1 Text) predominantly include correlation metrics between brightfield and various fluorescent channels, capturing spatial relationships between cellular components. Some of these mechanisms of action included compounds that were not individually distinguishable from negative controls, and CytoSummaryNet did not overcome the lack of phenotype in these cases. This suggests that while CytoSummaryNet excels in identifying certain cellular features, its effectiveness is limited when dealing with mechanisms of action that do not exhibit pronounced phenotypic changes.

## Discussion

This study proposes CytoSummaryNet, a Deep Sets-based model that automatically finds the optimal way to aggregate single-cell population profiles using self-supervised contrastive

learning. CytoSummaryNet learns an aggregation strategy by receiving single-cell morphological feature input data and providing the aggregated profile as an output. The model generally outperforms average profiling in both replicate retrieval and mechanism of action retrieval tasks, with the relative performance improvement depending on how CytoSummaryNet is applied.

For mechanism of action retrieval on the primary dataset (cpg0001), CytoSummaryNet achieves 36% to 60% better performance than average profiling on hold-out plates created with different experimental protocols than the training plates, provided that these hold-out plates are from the same stain set (e.g., Stain2, Stain3, etc.). A similar 30% improvement is observed on a second dataset (cpg0004 3.33 μM hold-out data). When applied to cpg0001 training and validation plates, CytoSummaryNet achieves 50% to 81% better performance than average profiling, which is affirmed by the 68% performance increase found for the cpg0004 training data. These results improve upon the ~20% gains previously observed using covariance features [13] albeit on a different dataset, and importantly, CytoSummaryNet effectively overcomes the challenge of recomputation after training, making it easier to use.

CytoSummaryNet does have difficulty generalizing to unseen test plates, both within the same stain set (Stain2-4) and in a different stain set (Stain5). The cpg0004 dataset, which tested the same compounds at two doses, provided insights into CytoSummaryNet's generalizability. While the model's performance on the unseen 3.33 μM dose point data was lower than on the seen 10 μM dose point data, it still improved upon average profiling, indicating that CytoSummaryNet can extract some general biological features. The model's tendency to amplify existing strong phenotypes and the presence of stronger phenotypic profiles at higher dose points may explain the larger increases in mAP for the 10 μM dose point data.

When trained on and applied to data within a given experimental condition and batch, CytoSummaryNet creates a better-organized feature space for discerning different compounds than average profiling. Still, not all compounds with the same mechanism of action cluster well, due to factors such as multiple annotated mechanisms, unannotated off-target effects, suboptimal cell type or dose, and technical artifacts [32].

CytoSummaryNet's learned aggregation method prioritizes the most informative cells, specifically those that are larger and more isolated from other cells. These cells are likely to provide clearer and more reliable information about a compound's effects, as phenotypes are more visible in these conditions. On the other hand, the least salient cells appear to be smaller, more crowded, and tend to contain spots of high-intensity pixels, which may indicate that they are dying, debris, or in some stage of cell division. These findings are consistent with recent studies that emphasize the importance of considering cell context when analyzing phenotypic variance and genetic interactions [33,34], supporting the notion that focusing on large, uncrowded cells can provide more accurate and informative data for understanding the effects of perturbations on cell populations.

CytoSummaryNet's performance is limited to data created under similar experimental conditions as the training data. While this requires retraining for each experiment, it is not a practical limitation because the strategy is self-supervised, requiring only that there be replicates of samples in the experiment for the training step.

Improving CytoSummaryNet's generalization capabilities, possibly by increasing training data variation [35], could eliminate the need for retraining. Recent studies exploring image-derived representations using self-supervised and self-supervised learning [35,36] could inspire future research on using learned embeddings instead of classical features to enhance model-aggregated profiles.

The architecture of CytoSummaryNet holds significant potential for broader applications beyond image-based cell profiling, accommodating tabular, permutation-invariant data and

enhancing downstream task performance when applied to processed population-level profiles. Its versatility makes it valuable for any omics measurements where downstream tasks depend on measuring similarity between profiles. Future research could also explore CytoSummary-Net's applicability to genetic perturbations, expanding its utility in functional genomics.

In conclusion, CytoSummaryNet provides an easier-to-apply and better-performing method for aggregating single-cell image feature data compared to previously published strategies and the average profiling baseline. The model likely achieves this by performing quality control, filtering out noisy cells. The improved mechanism of action retrieval metrics indicates that CytoSummaryNet's learned latent representation prioritizes biological signal over technical variance at both the cell and plate levels. Although CytoSummaryNet cannot be directly transferred to out-of-distribution data, it can be readily re-trained on new data, making it a valuable tool for improving results in future cell profiling studies.

## Methods

### Architecture

Our proposed model follows the general Deep Sets architecture [25]. The Deep Sets architecture can process permutation invariant data and learn to estimate first and second-order moments from the input data. Permutation invariance is necessary for aggregating sets of single-cell data into a sample profile, because the order in which the cells are processed should not influence the output.

After segmentation, thousands of features are measured for each cell, Fig 1B. These feature vectors are aggregated using a function $f(X)$, where $X \in \mathbb{R}^{M \times D}$ is the input set of single cell profiles, to get a single profile representing the cell population. There are multiple ways of defining the aggregation function $f(x)$. Our proposed architecture, Fig 1C, consists of two functions φ and ρ, which are simple fully connected neural networks, capable of approximating arbitrary polynomials. First, the model transforms the input set $X$ by transforming each single-cell profile (row) $x_m \in \mathbb{R}^D$ of the set using neural network φ: $\mathbb{R}^D \rightarrow \mathbb{R}^N$. φ consists of a single fully connected layer with 2048 nodes followed by a leaky ReLU activation layer. All of these nonlinear representations $\varphi(x_m)$ are summed, collapsing the cell dimension M: $\sum_{m=1}^{M} \varphi(x_m)$. The output $z \in \mathbb{R}^N$ is then processed by the projection network ρ: $\mathbb{R}^N \rightarrow \mathbb{R}^L$, which applies more nonlinear transformations to create a final representation $v$ in the loss space ($v = \rho(z)$). The function ρ consists of two subsequent fully connected layers of 512 and 2048 nodes respectively, both followed by leaky ReLU activations.

### Loss function

The model is trained using contrastive learning by computing the Supervised Contrastive (SupCon) loss [37]. To train contrastive learning models, one has to define positive and negative sample pairs. We define a positive pair as two samples perturbed with the same compound and a negative pair as two samples perturbed with different compounds. The SupCon loss is different from the commonly used SimCLR [38] in only one aspect: it takes into account all positive pairs of a certain sample instead of only one at a time, Eq 1. SupCon loss pulls positive pairs together in the embedding space, while simultaneously pushing apart negative pairs, Fig 1D. In this study, cosine similarity is used as the distance metric. The loss shows benefits for robustness to natural corruptions, hyperparameter settings, and inherently performs hard positive and hard negative mining when used in combination with cosine similarity [37].

$$L^{sup} = \sum_{i \leq I} \frac{-1}{|P(i)|} \sum_{p \in P(i)} log \frac{exp(v_i \cdot v_p / \tau)}{\sum_{a \in A(i)} exp(v_i \cdot v_a / \tau)} \qquad (1)$$

I: total number of samples
$\tau$: temperature constant (hyperparameter)
$P(i)$: *all positive samples for the current sample i*
$A(i)$: *all negative samples for the current sample i*
$v_i = \rho(\sum_{m=1}^{M} \varphi(x_m))$, where $x_m \in X_i$

## Model evaluation

After training the model, the projection network $\rho$ is often discarded, and the summed representation $z$ is used for downstream analysis instead [25]. However, $\rho$ is not discarded here because the projection it has learned is tied to the evaluation task. One of the main applications of image-based cell profiling is discovering the unknown mechanism of action of a certain compound. To that end, in addition to replicate retrieval (the training task), the proposed model is evaluated using mechanism of action retrieval. Mechanism of action retrieval is evaluated by quantifying a profile's ability to retrieve the profile of different compounds with the same annotated mechanism of action. If the model has learned to amplify the phenotypic signature of a sample's profile, finding other compounds with the same mechanism of action should also become easier.

The performance in mechanism of action retrieval and replicate retrieval (termed "phenotypic consistency" and phenotypic activity", respectively [26]) are compared between model-based profiling, which uses the learned aggregation, and average profiling. The performance is quantified by calculating the mean average precision (mAP) [26]. Average precision (AP) is an information retrieval metric that indicates whether the model can correctly identify all the positive examples without accidentally marking too many negative examples as positive. The calculation for the AP of a single query is shown in *Eq 2*. To compute the AP, a rank order of sample profiles is required. The top of the rank order corresponds to profiles most similar to the queried profile and vice versa. This rank order is created by calculating the cosine similarity between all $K$ sample profiles.

$$AP = \sum_{k=1}^{K} \left( r(k) - r(k-1) \right) p(k) \tag{2}$$

K: total number of sample profiles
$p(k)$: *precision at the k-th position of the rank order*
$r(k)$: *recall at the k-th position of the rank order*

A compound's AP is calculated by taking the average AP over all its replicate compound profiles, while a mechanism of action's AP additionally averages over the same mechanism of action compound profiles. The mAP is the mean AP over all of the compounds. If a compound does not have other compounds with the same mechanism of action, the compound and its replicates are left out of the mAP computation for mechanism of action retrieval.

## Interpretability analysis

Deep learning models are notoriously difficult to interpret. However, interrogating them has led to useful new insights in previous studies [39] and allows users to better understand the reasoning behind a model's output. In this study, a combination of sensitivity analysis (SA) [40] and a method similar to critical point analysis (CPA) used in the PointNet study [41] was used to investigate possible biological foundations for the model's output.

SA explains a model's prediction based on locally evaluated partial derivatives. The partial derivatives of a model's output $f(X)$ are calculated with respect to each entry in the input matrix $x_{m,d}$ by backpropagating the loss function $L^{sup}$. Afterward, the absolute value is taken of these partial derivatives, *Eq 3*. These values can then be summed over either the cell or feature dimensions to respectively get the feature or cell relevance scores. In this case, the relevance score per cell is computed. The analysis assumes that the most relevant input values are those to which the model's output is the most sensitive. Thus, inputs that receive a high relevance score will, when changed, make it more or less likely for the model to make a certain prediction. As a result, high relevance values can also characterize input patterns that the model would like to see removed to improve its performance for the predicted class. These patterns, e.g., noise, may not be linked to the class of interest.

$$R_{m,d} = ||\frac{\delta}{\delta x_{m,d}} f(X)|| \tag{3}$$

To counteract some of the potential noisy predictions of SA, CPA is used additionally for calculating the relevance scores. Since the model architecture is permutation invariant, each input cell vector is processed independently. In the PointNet study [41], CPA consisted of finding the input points with the maximum value for each feature. These points were found to form the skeleton of the input, meaning that they are the most relevant points for defining it. Their model's permutation invariant operation was a max pooling operation that inherently selected these points. In this study, a summation is used instead, although the reasoning is the same: cells with high activation values before the permutation invariant operation contribute more to the output of the model than those with low activation values. The CPA relevance score was calculated per cell by taking the L1-norm of their respective activations of the first fully connected layer.

The relevance scores of SA and CPA are min-max normalized per well and then combined by addition. The combination of the two is again min-max normalized, resulting in the combined relevance score. This score was used instead of the separate relevance scores because averaging was expected to cancel out some of the potentially noisy predictions. We show that the combined relevance score indeed achieves higher Pearson correlations with the CellProfiler features than the separate relevance scores, *Tables E-J in S1 Text*. Cells with the highest ($>0.8$) or lowest ($<0.2$) combined relevance scores are denoted as the 'most relevant' or 'least relevant' cells, respectively.

Multiple methods are used to investigate what the model has learned. First, the model and average-aggregated profiles of the top 15 same mechanism of action compound pairs, based on their mAP for mechanism of action retrieval, are visualized using a UMAP [31]. Four different values were tried for the UMAP hyperparameter *n_neighbors*, which balances local versus global structure in the data. From these UMAPs, one was chosen based on the best presentation of the clusters that were consistently visible throughout all UMAPs. The second method calculates the Pearson correlation between all of the input CellProfiler features and the combined relevance scores to get a better understanding of what features the model prioritizes. Finally, the most and least relevant cells are visualized and analyzed in the raw microscopy images to link the model's output to the underlying cell biology of the compound perturbations. Plates from multiple StainX subsets are used for these analyses to find commonalities between them.

## Experimental setup

### Data

We tested this method separately on the *cpg0001-cellpainting-protocol* dataset [42] (abbreviated to cpg0001 here), from the JUMP consortium [43] and batch 1 of the *cpg0004-lincs* dataset [32] (abbreviated to cpg0004 here). Both datasets are available from the Cell Painting Gallery [27] on the Registry of Open Data on AWS https://registry.opendata.aws/cellpainting-gallery/.

The 384-well plates that make up the cpg0001 dataset each contain four replicates of each of the 90 different compound perturbations. This dataset was created with the aim of optimizing the analysis pipeline for image-based cell profiling, resulting in a lot of technical variation. The analysis pipeline varied in dye concentration, cell permeabilization, cell seeding, exposure, pixel binning, compound dose, or microscopy method (confocal versus widefield). Each well was seeded with U2OS cells (a bone cancer cell line) in the solvent dimethyl sulfoxide (DMSO). Each well was either perturbed with one of 90 compounds or used as a negative control using only the solvent. The well position of each compound perturbation was fixed, i.e., the same plate layout was used across all plates. In total, 42 plates from four different optimization experiments called Stain2, Stain3, Stain4, and Stain5 were used. Unlike plates within an experiment, these were carried out at different times and therefore introduce additional technical variance, e.g., due to changes in laboratory conditions, sample manipulation, or instrument calibration.

The cpg0004 dataset (Batch 1) was created with a single analysis pipeline, however, it contains much more biological variation due to the 1.258 different compound perturbations used. These datasets used 384-well plates. The cells were stained using the Cell Painting protocol [28], after which images were taken with a microscope. The dataset used A549 cells (an adenocarcinomic cell line) in a DMSO solvent. The wells on each plate were perturbed with 56 different compounds in six different doses. Every compound was replicated 4 times per dose, with each replicate on a different plate. This requires the model to find replicates across plates during training instead of within plates like in the cpg0001 dataset. In this study, only the highest and second-highest dose points are used: 10 μM and 3.33 μM. 28 different plate layouts were used across all 136 plates.

A subset of the data was used for evaluating the mechanism of action retrieval task, focusing exclusively on compounds that belong to the same mechanism class. The Stain plates contained 47 unique mechanisms of action, with each compound replicated four times. Four mechanisms had only a single compound; the four mechanisms (and corresponding compounds) were excluded, resulting in 43 unique mechanisms used for evaluation. In the LINCS dataset, there were 1436 different mechanisms, but only 661 were used for evaluation because the remaining had only one compound.

### Preprocessing

After the cells were segmented in each image, features were extracted using CellProfiler [44]. A subset of 1324 features was taken which were available in all plates of cpg0001, because CytoSummaryNet requires a fixed number of input features and to reduce the model's computational burden. Similarly, a subset of 1745 features was taken from the plates in cpg0004. The complete list of these feature names can be found in the GitHub repository of this project. Rows containing any NaNs were removed. The data used to compute the average profiles and train the model were standardized at the plate-level, ensuring that all cell features across the plate had a zero mean and unit variance. The negative control wells were then removed from all plates.

A commonly used pipeline was employed to calculate the average-aggregated profiles [21]. Only the selected subset of features was used to allow for a fair comparison with the model-aggregated profiles. After calculating the average-aggregated profile for each well in a certain plate, the features were RobustMAD normalized by subtracting their median and dividing by their mean absolute deviation. The final average profiles were acquired by applying feature selection using a variance and correlation threshold. The full data processing workflows are available at https://github.com/jump-cellpainting/pilot-data-public (for cpg0001) and https://github.com/broadinstitute/lincs-cell-painting (for cpg0004). For the model, the outputs were used directly as the aggregated profiles.

## Diversity of stain sets

Stain2-5 comprise a series of experiments which were conducted sequentially to optimize the experimental conditions for image-based cell profiling. These experiments gradually converged on the most optimal set of conditions; however, within each experiment, there were significant variations in the assay across plates. To illustrate the diversity in experimental setups within the dataset, we will highlight the differences between Stain2 and Stain5.

Stain2 encompasses a wide range of nine different experimental protocols, employing various imaging techniques such as Widefield and Confocal microscopy, as well as specialized conditions like multiplane imaging and specific stains like MitoTracker Orange. This subset also includes plates acquired with strong pixel binning instead of default imaging and plates with varying concentrations of dyes like Hoechst. As a result, Stain2 exhibits greater variance in the experimental conditions across different plates compared to Stain5.

In contrast, Stain5, the last experiment in the series, follows a more systematic approach, consistently using either confocal or default imaging across three well-defined conditions. Each condition in Stain5 utilizes a lower cell density of 1,000 cells per well compared to Stain2's 2,500 cells per well. Being the final experiment in the series, Stain5 had the least variance in experimental conditions.

For training the models, we typically select the data containing the most variance to capture the broadest range of experimental variation. Therefore, we chose Stain2-4 for training, as they represented the majority of the data and captured the most experimental variation. We reserved Stain5 for testing to evaluate the model's ability to generalize to new experimental conditions with less variance.

All StainX experiments were acquired in different passes, which may introduce additional batch effects.

## Data augmentation

Data augmentation was used to increase the generalization of the model to unseen samples. During training, for each batch, a number of cells was randomly sampled with replacement from every well. The number of cells that was sampled was itself a number sampled from a Gaussian distribution. Although they remained mostly similar, this created unique sets of cells for each compound in every batch. The number of sets of cells that was sampled for each compound in each epoch is a tunable hyperparameter. Each sampled set could consist of cells from a single well or a combination of two at random. This decision was based on a coin flip, which was performed for each sample. This last form of augmentation should help decrease plate-layout effects, which introduce a technical bias into the single-cell feature data based on which well position the population is in.

The model training process consists of many tunable hyperparameters, which were chosen using a random search. The AdamW optimizer [45] is used with a learning rate of $5 \times 10^{-4}$ and

weight decay $10^{-2}$. The model is trained for 100 epochs or until the best mAP is achieved on the validation compounds of the training and validation plates. An overview of all the tunable parameters, including the data augmentation parameters, is given in *Table D in S1 Text*.

## Supporting information

**S1 Text. Fig A: Additional results for testing the generalizability of CytoSummaryNet.** Fig B: Estimating second- and third-order moments using Deep Sets and contrastive learning. Two-dimensional point set distributions for ten different population classes. Fig C: Estimating second- and third-order moments using Deep Sets and contrastive learning. Effect of sphering on populations. Fig D: Estimating second- and third-order moments using Deep Sets and contrastive learning. Ten populations with varying skewness. Fig E: Measuring plate similarity with a hierarchical clustermap. Fig F: Additional cell image field of views 1. Fig G: Additional cell image field of views 2. Fig H: Layout of training and validation compounds. Fig I: Relevant features for CytoSummaryNet improvement. Fig J: Number of predictable mechanisms of action across different mAP thresholds Fig K: Investigating CytoSummaryNet's cell prioritization. Table A: Estimating second- and third-order moments using Deep Sets and contrastive learning. Experiment results in mAP actual data. Table B: Estimating second- and third-order moments using Deep Sets and contrastive learning. Experiment results toy dataset. Table C: Training, validation, and test set stratification. Table D: CytoSummaryNet hyperparameter overview. Table E: Relevance correlation with CellProfiler features. Additional results SA and CPA negative correlation. Table F: Relevance correlation with CellProfiler features. Additional results SA and CPA positive correlation. Table G: Relevance correlation with CellProfiler features. Additional results SA negative correlation. Table H: Relevance correlation with CellProfiler features. Additional results SA positive correlation. Table I: Relevance correlation with CellProfiler features. Additional results CPA negative correlation. Table J: Relevance correlation with CellProfiler features. Additional results CPA positive correlation. Table K: List of features that most strongly differentiate average profiles from negative controls, for a subset of mechanisms of action (MoA). Table L: Computation time and storage requirements. (PDF)

## Acknowledgments

The authors appreciate helpful comments from Juan Caicedo, Alexandr Kalinin, Alán Muñoz González, Niranj Chandrasekaran, Srijit Seal, Ellen Su, Beth Cimini, and Erin Weisbart. The authors also thank Josien Pluim for her mentorship during RvD's internship.

## Author Contributions

**Conceptualization:** Mehrtash Babadi, Anne E. Carpenter, Shantanu Singh.

**Data curation:** Robert van Dijk.

**Formal analysis:** Robert van Dijk, John Arevalo.

**Funding acquisition:** Anne E. Carpenter.

**Investigation:** Robert van Dijk, John Arevalo, Shantanu Singh.

**Methodology:** Robert van Dijk, John Arevalo, Shantanu Singh.

**Project administration:** Shantanu Singh.

**Resources:** Anne E. Carpenter.

**Software:** Robert van Dijk.

**Supervision:** Shantanu Singh.

**Validation:** Robert van Dijk, Shantanu Singh.

**Visualization:** Robert van Dijk.

**Writing – original draft:** Robert van Dijk.

**Writing – review & editing:** Robert van Dijk, John Arevalo, Mehrtash Babadi, Anne E. Carpenter, Shantanu Singh.

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
