## [Decision Letter · Decision Letter 0]

10 Oct 2024

Dear Dr Singh,

We are pleased to inform you that your manuscript 'Capturing cell heterogeneity in representations of cell populations for image-based profiling using contrastive learning' has been provisionally accepted for publication in PLOS Computational Biology.

Best regards,

Virginie Uhlmann

Academic Editor

PLOS Computational Biology

Marc Birtwistle

Section Editor

PLOS Computational Biology

Reviewer's Responses to Questions

**Comments to the Authors:**

Reviewer #1: Dear Authors,

In my eyes, you have successfully addressed all my comments. I do believe that the manuscript gained a lot in clarity and openness. And as such will be useful to a wider audience. Again, it might be interesting to see how the model would perform on the JUMP-CP CRISPR data for retrieval of functionally related genes. Yet, this is even more challenging as cell-to-cell heterogeneity plays a larger role. However, this might be done in another manuscript.

Best,

Reviewer 3

Reviewer #2: The authors have addressed my concerns. In my opinion, the manuscript is ready for publication and will be a nice contribution to the field.

There is one last issue that I recommend but should be decided whether to include by the authors. Fig. K1 in the Supplementary Material shows an “application appropriate” measurement (PMID: 37433995) in the number of predictable MoAs as a function of an increasing mAP threshold. In my opinion this is the direct measurement that matters the most in the context of MoA prediction. Thus I highly recommend the following: (1) move it to the main text, (2) calculate the improvement in percentage according to this measurement (and report their range in the abstract instead of the mPA %), (3) include a direct comparison to the contribution of contrastive learning alone, shown in Table 1 “mAP improvement (population average)” (nice to have).

**Have the authors made all data and (if applicable) computational code underlying the findings in their manuscript fully available?**

Reviewer #1: Yes

Reviewer #2: Yes

PLOS authors have the option to publish the peer review history of their article (what does this mean?). If published, this will include your full peer review and any attached files.

Reviewer #1: No

Reviewer #2: **Yes: **Alon Shpigler and Assaf Zaritsky

---

## [Editor Report · Acceptance letter]

31 Oct 2024

PCOMPBIOL-D-24-01350 

Capturing cell heterogeneity in representations of cell populations for image-based profiling using contrastive learning

Dear Dr Singh,

I am pleased to inform you that your manuscript has been formally accepted for publication in PLOS Computational Biology. Your manuscript is now with our production department and you will be notified of the publication date in due course.

With kind regards,

Dorothy Lannert
